# Photo-excited charge carriers suppress sub-terahertz phonon mode in silicon at room temperature

Bolin Liao[1,†], A.A. Maznev[2], Keith A. Nelson[2] & Gang Chen[1]

There is a growing interest in the mode-by-mode understanding of electron and phonon transport for improving energy conversion technologies, such as thermoelectrics and photovoltaics. Whereas remarkable progress has been made in probing phonon–phonon interactions, it has been a challenge to directly measure electron–phonon interactions at the single-mode level, especially their effect on phonon transport above cryogenic temperatures. Here we use three-pulse photoacoustic spectroscopy to investigate the damping of a single sub-terahertz coherent phonon mode by free charge carriers in silicon at room temperature. Building on conventional pump–probe photoacoustic spectroscopy, we introduce an additional laser pulse to optically generate charge carriers, and carefully design temporal sequence of the three pulses to unambiguously quantify the scattering rate of a single-phonon mode due to the electron–phonon interaction. Our results confirm predictions from first-principles simulations and indicate the importance of the often-neglected effect of electron–phonon interaction on phonon transport in doped semiconductors.

[1] Department of Mechanical Engineering, Massachusetts Institute of Technology, Cambridge, Massachusetts 02139, USA. [2] Department of Chemistry, Massachusetts Institute of Technology, Cambridge, Massachusetts 02139, USA. † Present address: Division of Chemistry and Chemical Engineering, California Institute of Technology, Pasadena, California 91125, USA. Correspondence and requests for materials should be addressed to K.A.N. (email: kanelson@mit.edu) or to G.C. (email: gchen2@mit.edu).

The ability to understand microscopic transport and interaction of energy carriers 'mode by mode' holds promise of reshaping energy–material research at the most fundamental level. First-principles simulations partially provide this ability[1–3], although they have hitherto been limited to relatively simple material systems. On the experimental side, recent developments in ultrafast photoacoustic spectroscopy[4–6], inelastic neutron scattering[7] and quasiballistic phonon mean-free-path spectroscopy[8–10] enabled progress in probing phonon–phonon interaction strength of individual phonon modes or the distribution of phonon modes with respect to phonon–phonon-interaction-limited mean free paths. These tools can provide guidance for designing nanostructured thermoelectric materials[11]. The same level of insight is also desirable for electron–phonon interaction, which is among the most important interactions of (quasi)particles in condensed matter physics and material science. Electron–phonon interaction is the major contributor to electrical resistance in most inorganic metals and semiconductors[12] above cryogenic temperatures, plays the central role in the microscopic theory of superconductivity[13] and forms the basis of polaron formation and transport in conjugated polymers[14]. Given its paramount importance, numerous experimental techniques have been developed to probe the electron–phonon interaction in various materials directly or indirectly, with most of them examining the effect of electron–phonon interaction on electrons. For example, the collective effect of the interactions among all phonons and electrons that participate in transport can be inferred from electrical transport experiments[12,15]. Alternatively, the average electron–phonon coupling strength can be directly measured by investigating the timescale of equilibration of electrons and phonons in ultrafast optical pump–probe experiments[16,17]. In addition, the angle-resolved photoemission spectroscopy can directly map out the electronic band structure near the material surface, and the linewidths of the electronic states provide specific information of the interaction strength between a single-electron state with all phonon modes[18,19].

On the other hand, the effect of electron–phonon interaction on phonons has been less studied, both theoretically and experimentally. Previous measurements on metals usually apply a high magnetic field to 'freeze out' the electrons and then measure the change of thermal conductivity[20]. In most cases the change is small due to the small energy scale of magnetic fields, and most measurements were done at cryogenic temperatures. An alternative way to probe the phonon-specific information of electron–phonon interaction in metals is through superconducting tunnelling spectroscopy[21]. From superconducting tunnelling spectroscopy the Éliashberg function[22] $\alpha^2 F(\omega)$ can be extracted, which reflects the interaction strength of electrons near the Fermi surface with phonons with a specific frequency $\omega$. However, it is limited to superconductors and cannot resolve individual phonon modes. Similarly, inelastic neutron scattering was used to measure and compare phonon linewidths of metals in the normal and superconducting states[23], the difference between which gives the phonon damping due to electron–phonon interaction in the normal state. The change of phonon damping in metals across the superconductor transition has also been studied by ultrasonic attenuation experiments[24–27] in the megahertz frequency range.

Early experiments on semiconductors mostly focused on the effect of carriers introduced by doping on the thermal conductivity[28,29]. One difficulty of these experiments is to separate the contributions to phonon scattering from carriers themselves and from the impurities introduced by doping. The same difficulty stands in inelastic neutron-scattering measurements[30] and ultrasonic attenuation experiments[31] of doped semiconductors. An alternative way of introducing carriers

is by electrostatic gating. Owing to the short screening length (typically a few nanometres in semiconductors), the induced carriers are confined in a thin layer and cannot effectively scatter phonons because of the short interaction time. And thus, measurements of phonon damping due to electrostatically induced carriers have only been carried out at cryogenic temperatures[32,33].

The recent advancement of thermoelectrics has revived the interest to the effect of the electron–phonon interaction on phonon transport, since most thermoelectric materials are heavily doped semiconductors with carrier concentrations in the range of $10^{19}$–$10^{21}$ cm$^{-3}$, and the electron–phonon interaction can potentially be an important factor in scattering phonons in this regime. Liao et al.[34] have shown through first-principles calculations that the lattice thermal conductivity of silicon at room temperature can be reduced by 45% due to the electron–phonon interaction at a carrier concentration of $10^{21}$ cm$^{-3}$. Furthermore, the calculations in ref. 34 have resolved the effect of electron–phonon interaction on each individual phonon mode. However, an experimental verification of these findings has been lacking.

In this article, we use a three-pulse femtosecond photoacoustic technique to quantify the effect of the electron–phonon interaction on the lifetime of a single-coherent phonon mode in a silicon membrane at room temperature and achieve good agreement between experimental results and first-principles calculations for the phonon lifetime as a function of the carrier concentration. Building on conventional femtosecond photoacoustic spectroscopy[4,5,35], we introduce an extra excitation beam to optically generate electron–hole pairs, and carefully design the temporal sequence of the three pulses to add extra damping due to electron–phonon interaction to the second acoustic arrival (the echo of the first pulse after a round trip inside the membrane) while not affecting the first arrival. By comparing the magnitudes of the first and the second arrivals of the acoustic pulse, the effect of electron–phonon interaction on phonon damping can be unambiguously quantified. With this design, we completely rule out the effect of phonon-impurity scattering, as free charge carriers are introduced optically rather than by doping. Furthermore, we optically excite carriers uniformly through the thickness of the sample, so that the phonon mode has enough time to interact with the carriers, which allows a sufficiently strong effect to be measured at room temperature. Thus, our method overcomes the previously stated difficulties that have prevented direct quantification of the effect of electron–phonon interaction on phonon transport, and our measurement results show good agreement with the first-principles calculations of ref. 34. In particular, we find that beyond a carrier concentration of $2 \times 10^{19}$ cm$^{-3}$, the electron–phonon scattering provides the dominant mechanism of the phonon decay.

## Results

**Experimental set-up.** In conventional ultrafast photoacoustic spectroscopy[4,6,35,36], an acoustic strain pulse is launched in a thin sample by an ultrafast optical pump pulse. This optical pump pulse is absorbed through electronic transitions, and then an acoustic strain pulse is released by relaxation of the strain generated either by photo-excited carriers or thermal expansion. Subsequently, this acoustic strain pulse travels back and forth inside the thin sample and is recorded by an ultrafast optical probe pulse. The optical response in the probe pulse is generated through the photoelastic response of the material to the strain pulse[35]. If the probe penetration depth is relatively large (on the order of the wavelength in the medium or larger), the

transient reflectivity signal is typically dominated by high-frequency Brillouin oscillations, resulting from the interference of light reflected from the sample surface and from the strain pulse[36–38]. This process is illustrated in Fig. 1b. The Brillouin oscillations typically lead to a narrowband signal with a peak frequency (Brillouin frequency $f_B$) determined by the longitudinal sound speed $v_L$, the refractive index $n$ of the sample and the probe wavelength $\lambda$ as $f_B = (2nv_L)/\lambda$. In silicon with a 390 nm probe beam, the Brillouin frequency is $\sim 250$ GHz. By comparing the amplitudes of the frequency components of the acoustic pulse at different propagation lengths, the acoustic damping due to losses during the propagation inside the membrane can be quantified. A typical signal of the first two echoes of the acoustic pulse (hereafter we refer to the recorded arrivals of the acoustic pulse at the specific sample surface as 'echoes', although technically the first arrival is not an echo) is shown in Fig. 1c. In the conventional pump–probe set-up, the phonon damping is caused by phonon–phonon interaction in the bulk of the sample and scattering by surface roughness[4,36].

To measure the phonon damping caused by free charge carriers, we introduce another optical pulse to generate carriers inside the sample. We choose to use 780 nm optical pulses with a

1.7 µm-thick silicon membrane sample. At 780 nm, the optical penetration depth of silicon is $\sim 8$ µm, so that the carriers are generated nearly uniformly within the membrane. For pump and probe pulses, we choose 390 nm wavelength, at which the penetration depth is only $\sim 60$ nm.

Key components of the experimental set-up are illustrated in Fig. 1a. Details of the set-up are described in Methods. Briefly, we focus three pulsed laser beams on the same spot of a silicon membrane: 390 nm pump; 390 nm probe; and an additional 780 nm excitation beam to generate carriers. Since we use a pulsed laser to generate carriers, the timing of the three pulses is crucial. We design the pulse sequence such that the 780 nm excitation pulse arrives right after the first echo is recorded by the 390 nm probe, marked by the arrow in Fig. 1c. At delay time $t = 0$ ps, the 390 nm pump pulse hits the front side of the sample, and launches an acoustic strain pulse. It takes $\sim 210$ ps for the strain pulse to traverse the 1.7 µm-thick membrane and be recorded by the 390 nm probe beam on the backside of the sample. After that, this acoustic strain pulse is reflected from the sample surface and starts the second round trip inside the membrane. In our design, the 780 nm pulse arrives at $t = 260$ ps, right after the first arrival of the acoustic pulse is recorded. The 780 nm pulse generates free electron–hole pairs uniformly inside the membrane. Since silicon is an indirect-gap semiconductor, the recombination time is relatively long. Therefore, the generated electron–hole pairs remain in the membrane for up to a few nanoseconds, and will damp the acoustic strain pulse during its second round trip. At $t = 620$ ps, the second echo is recorded by the probe beam when the strain pulse reaches the backside of the membrane again. The decay of the phonon mode at the Brillouin frequency (250 GHz) can now be quantified by comparing the magnitude of spectral peaks at 250 GHz in Fourier transforms of the first and second echoes. Furthermore, the contribution to the phonon damping from the photo-excited carriers can be isolated from phonon–phonon scattering and boundary loss by comparing the total attenuation with and without the 780 nm excitation beam. One important advantage of this design is that only the ratio, not the absolute amplitudes, of the two pulses matters, so that the fluctuations of the laser power and pointing on longer timescales do not affect the measurement. Technical considerations of the material heating caused by the excitation beam and the carriers generated by the 390 nm pump beam are given in Discussion.

Figure 2 shows the response of the silicon membrane to the 780 nm excitation beam alone, without the 390 nm pump beam. One can see that higher power leads to larger change in the reflectance signal at 260 ps when the excitation pulse arrives, indicating a higher carrier concentration. The decay of the signal with time indicates the carrier recombination process. The recombination process is faster for higher carrier concentration, as expected, but the lifetime is still sufficiently long to ensure that the carrier concentration does not drop significantly within the measurement window (indicated by the dashed lines, 260–620 ps). The oscillations on the decay curves represent the lowest-order thickness resonance of the membrane at 2.5 GHz, corresponding to the $\sim 390$ ps acoustic round-trip time, which is far separated from the 250 GHz Brillouin oscillation. The decay curves drop below the baseline for higher excitation power due to the reflectance change of silicon caused by the slight temperature rise. It is known that the strain caused by the carrier generation and by heating in silicon has opposite signs[39].

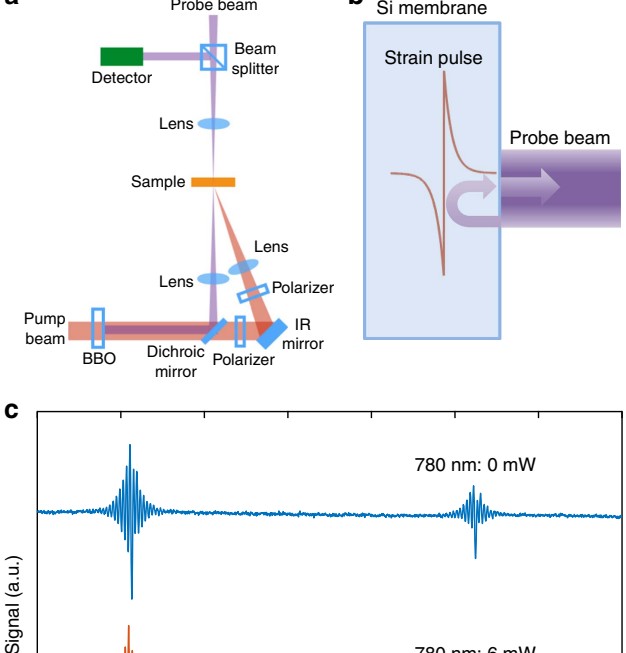

**Figure 1 | Overview of the experiment.** (**a**) Schematic of the experimental set-up. Not shown are components of the set-up standard for ultrafast optical pump–probe experiments such as the laser, the optical delay line and so on. (**b**) Schematic showing the strain pulse generated by the pump pulse in a silicon membrane, and the generation of Brillouin oscillations: interference of probe light reflected from the sample surface and the strain pulse. (**c**) Typical signal (the first two echoes) without the 780 nm excitation beam and with 6 mW of 780 nm excitation. The arrow marks the time delay at which the 780 nm excitation pulse arrives. The slow transient after the arrival of the 780 nm excitation is due to the carrier generation. IR, infrared; BBO, bismuth borate.

**Measurement and data analysis.** Figure 1c shows the first and second acoustic pulses with both the 390 nm pump beam and 6 mW of 780 nm excitation beam. The slower transient due

to carrier generation can be observed after the arrival of the 780 nm excitation pulse. It is clearly seen that the fast Brillouin oscillations in the second echo are suppressed by the 780 nm excitation beam, a clear evidence of phonon damping by the photo-excited electron–hole pairs, whereas those in the first echo are not affected by the presence of the 780 nm excitation beam, ruling out the effect of steady-state temperature rise on phonon damping. Figure 3a shows the recorded first and second echoes for different powers of the 780 nm excitation. One can see that the fast Brillouin oscillations in the second echo get increasingly suppressed and eventually disappear at higher 780 nm powers.

To quantify the phonon lifetime due to electron–phonon interaction, Fourier transforms of the first and second echoes are

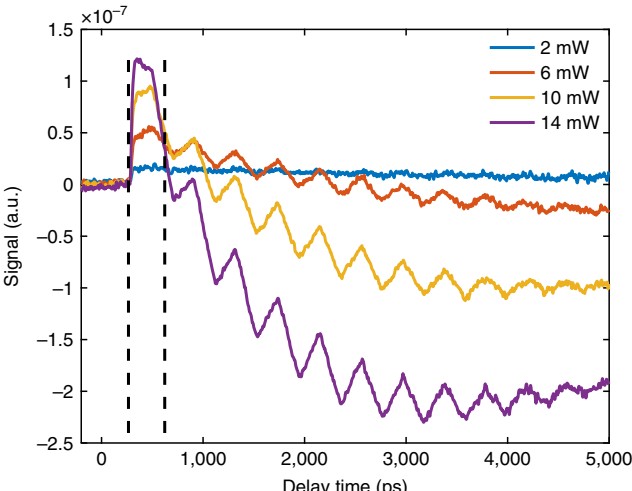

**Figure 2 | The response with only the 780 nm excitation beam.** The 780 nm excitation power is varied from 6 to 14 mW. The responses are caused by carrier generation, with different carrier concentrations and lifetimes at different excitation power. Higher 780 nm power leads to larger jump of the signal at 260 ps, indicating that higher carrier concentration is generated. When the generated carrier concentration is higher, the recombination process is faster, reflected by the faster decay of the signal. At longer delay times, the signal drops below the baseline, as a result of the slight temperature rise. The superposed oscillation is the lowest-order thickness resonance of the membrane at 2.5 GHz.

carried out as shown in Fig. 3b. As expected, the amplitude of the frequency component at 250 GHz of the second echo is clearly suppressed by the 780 nm excitation, even down to the noise level when 14 mW of 780 nm excitation beam is used. Measurements beyond this power are thus not possible.

To calculate the 250 GHz phonon-scattering rate (reciprocal of the lifetime) due to the electron–phonon interaction, we separate the additional contribution of phonon attenuation from electron–phonon interaction by comparing the overall attenuations with and without the excitation beam. The details of the calculation, and the estimation of carrier concentration generated by the 780 nm excitation pulse are given in Methods. To eliminate the error caused by laser fluctuations, we did reference measurements without the 780 nm excitation before and after each measurement with the 780 nm excitation, and repeated the measurements multiple times.

The 250 GHz phonon-scattering rate due to the electron–phonon interaction is plotted in Fig. 4. The reported data were measured on five different locations of the membrane, and each data point represented average of 10–30 measurements. The error bars are the s.d.'s of the measurement results that are averaged. Also plotted in Fig. 4 is the theoretical prediction based on the following equation derived from Fermi's golden rule[34]:

$$\frac{1}{\tau_{\mathbf{q}\nu}^{\text{ep}}} = \frac{(2\pi m^*)^{1/2} D_{\text{A}}^2}{(k_{\text{B}}T)^{3/2} g_{\text{d}}\rho \nu_{\text{s}}} \exp\left(-\frac{m^* \nu_{\text{s}}^2}{2k_{\text{B}}T}\right) n\omega_{\mathbf{q}\nu}, \qquad (1)$$

where $\tau_{\mathbf{q}\nu}^{\text{ep}}$ is the lifetime due to electron–phonon interaction of a specific phonon with wavevector $\mathbf{q}$ and branch index $\nu$, $m^*$ the density-of-state effective mass of the carriers, $D_{\text{A}}$ the acoustic deformation potential, $\rho$ the mass density of the material, $g_{\text{d}}$ the number of equivalent carrier pockets, $\nu_{\text{s}}$ the sound velocity, $n$ the carrier concentration and $\omega_{\mathbf{q}\nu}$ the angular frequency of the phonon mode. A similar expression for longitudinal optical phonons is also given in ref. 34. These expressions can be directly used in Callaway-type models for estimating the thermal conductivity. The contributions from the electrons and holes are calculated separately and directly summed with the assumption that the phonon scattering due to electrons and holes is independent. This assumption is valid in silicon at room temperature given its low exciton-binding energy[40]. The deformation potential values we used in this work are the acoustic deformation potential for electrons $D_{\text{A,e}} = 5.2$ eV and for holes $D_{\text{A,h}} = 4.8$ eV. With these values, equation (1) agrees with

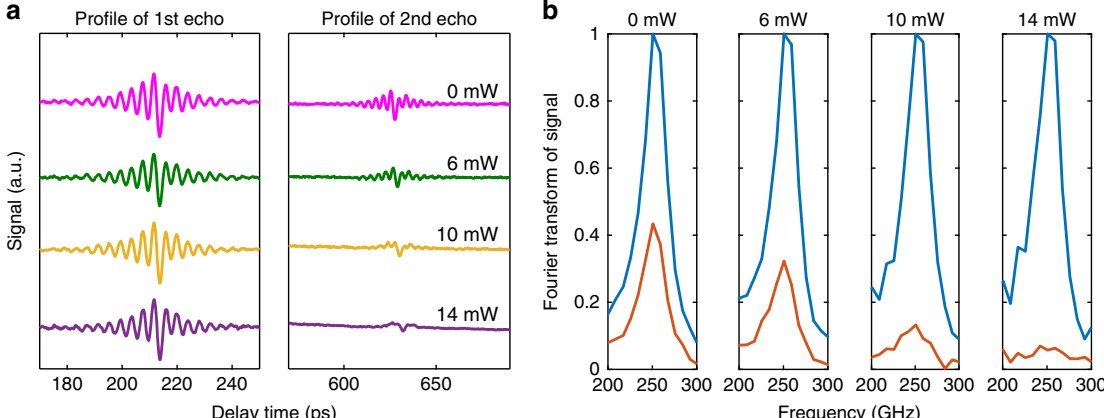

**Figure 3 | The attenuation of the second acoustic echo due to photo-excited carriers.** (**a**) The recorded profiles of the first and second echoes versus the power of the 780 nm excitation beam. The fast Brillouin oscillations in the second echo are suppressed by 780 nm excitation beam, while the first echo is not affected. (**b**) Fourier spectra of the first and second echoes versus the power of the 780 nm excitation beam. The blue lines are the spectra of the first echo, and the orange lines are the spectra of the second echo.

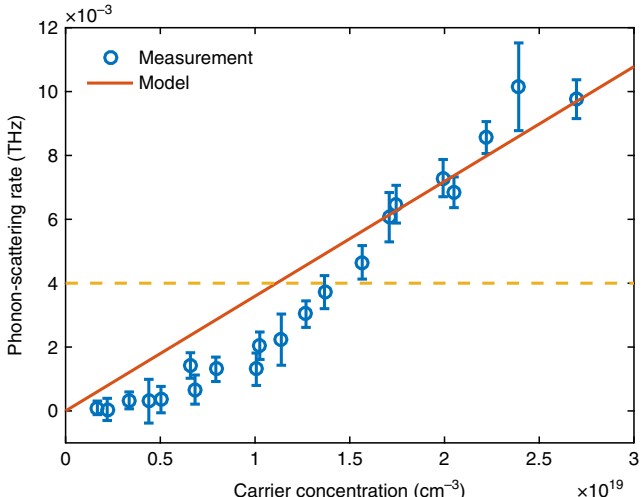

**Figure 4 | Comparison between experiment and theory.** The measured 250 GHz phonon-scattering rate due to electron–phonon interaction, compared with the theoretical prediction in ref. 34. The yellow dashed line labels the phonon attenuation level without the 780 nm excitation beam, as a reference. The reported data were measured on five different locations of the membrane, and each data point represented average of 10–30 measurements. The error bars are the s.d.'s of the measurement results that are averaged.

first-principles simulations in ref. 34 in the small wavevector regime. (The values given in ref. 34 were underestimated by a factor of two due to a missing numerical factor when fitting the simulation results. This error does not affect the simulation result itself.) Given the approximations made in data processing and possible errors in estimating the carrier concentrations (see Methods), the agreement between experiment and theory is reasonable.

## Discussion

There are several technical considerations. First, the heating caused by the 780 nm beam needs to be minimized. Given the single-pulse energy of 80 nJ at 10 mW, and the volumetric specific heat of silicon $\sim 1.6 \times 10^6 \, \text{JK}^{-1} \, \text{m}^{-3}$, the instantaneous temperature rise due to absorption of a 780 nm pulse can be estimated to be $\sim 4 \, \text{K}$. This is significantly smaller than the Debye temperature of silicon (645 K) and thus has negligible effect on phonon damping. The limited heat dissipation capability, however, of the silicon membrane also leads to an accumulated steady-state temperature rise. We estimate this temperature rise in a COMSOL simulation to be $\sim 15 \, \text{K}$ at the maximum power ($\sim 15 \, \text{mW}$) we use in the experiment (Supplementary Fig. 1). This temperature rise is still not expected to have observable effect on phonon damping, which is also verified by the experimental fact that the amplitude of the first echo is not affected by the presence of the 780 nm excitation (Fig. 3a).

Second, the 390 nm pump pulse also generates carriers. We note that Pascual-Winter *et al.*[41] suggested that the carriers generated by the pump could affect the measured intrinsic phonon lifetime in a GaAs-AlAs superlattice at 15 K, although room temperature measurements on the same material system[5] did not show this effect. In our case the contribution of the carriers generated by the 390 nm pump to the phonon damping is subtracted out when we compare the total damping with and without the 780 nm excitation. However, we do not want the carriers generated by the 390 nm pump to dominate those

generated by the 780 nm excitation. The typical power of the 390 nm pump beam we use in the experiment is 13 mW, $\sim 60\%$ of which is reflected by the membrane. Owing to the shallow absorption length ($\sim 60 \, \text{nm}$) at 390 nm, an extremely high ($\sim 5 \times 10^{20} \, \text{cm}^{-3}$) carrier concentration is initially generated within the thin 60 nm layer. At this carrier concentration, Auger recombination is strong and the recombination time is below 100 ps (ref. 42). Therefore, the carrier concentration will drop significantly before the acoustic pulse reaches the backside of the membrane for the first time, and further reduced by the diffusion of the carriers. With an average hole diffusivity of $10 \, \text{cm}^2 \, \text{s}^{-1}$ in silicon (ref. 43) (electron diffusivity is $\sim 30 \, \text{cm}^2 \, \text{s}^{-1}$; here we use the hole value because holes are more effective in scattering phonons[34]), the carriers will diffuse out to a layer of $\sim 300 \, \text{nm}$ in 200 ps, and given a recombination time of 100 ps, the carrier concentration is estimated to drop below $1 \times 10^{18} \, \text{cm}^{-3}$. To test this conclusion, we measured the 250 GHz phonon lifetime without the 780 nm excitation beam with different 390 nm pump power from 6 to 15 mW at the same location on the membrane, and did not see any systematic reduction of the phonon lifetime with increasing pump power at 390 nm (Supplementary Table 1).

It should be noted that although in our experiment the effect of the carriers produced by the pump pulse used to generate coherent phonons is found to be negligible, it may not be negligible in other experiments, where measurements of the acoustic phonon damping in semiconductors are accompanied by the photogeneration of free carriers. For example, Cuffe *et al.*[44] studied the lifetimes of the lowest-order dilatational thickness resonances in ultrathin silicon membranes using tightly focused (1.75 μm spot size) 800 nm pump pulses and they found that the measured lifetime is shorter than model predictions including phonon–phonon and phonon–boundary scatterings. Our results suggest that the additional phonon damping due to carriers generated by the pump may have contributed to their measurements.

It can be seen in Fig. 4 that the measured scattering rate is lower than the theoretical prediction for lower carrier concentrations. We note that the photo-excited electrons also scatter off thermal phonons. The lifetime (not recombination time) of electrons near the band edges due to electron–phonon interaction is $\sim 100 \, \text{fs}$ according to previous first-principles calculations[45], which is significantly shorter than the period of the phonon mode studied (4 ps). When applying Fermi's golden rule, it is implicitly assumed that the participating particles maintain their phases throughout the scattering process. In this case the electrons do not maintain their phases for a sufficiently long time (the phonon period) due to other scattering processes, and strictly speaking, this assumption of Fermi's golden rule is not fully satisfied. In the opposite limiting case, when the electron lifetime is much shorter than a phonon period, phonon damping due to free carriers can be analysed[31] by considering the relaxation of the carrier distribution in the strain field imposed by the phonon, the approach originally due to Akhiezer[46]. This relaxation-type theory[31] also predicts a linear dependence of the phonon decay rate on the carrier concentration. We note, however, that Akhiezer's approach requires the electron mean free path to be much smaller than the acoustic phonon wavelength, a requirement that is not quite met in our case, with the acoustic wavelength being as small as 34 nm. A comprehensive theory that would bridge Akhiezer's approach with that based on Fermi's golden rule[34] is currently lacking and will be part of our future pursuit. However, the fact that experimental results come close to the predictions of ref. 34 indicate that the latter approach may yield reasonable results outside the domain of its formal validity.

In summary, we have measured the lifetime of 250 GHz coherent longitudinal acoustic phonons due to scattering by photo-excited electrons and holes at room temperature. We have found that at carrier concentration beyond $2 \times 10^{19}$ cm$^{-3}$, scattering by charge carriers provides a dominant channel for the phonon decay. Our measurement lends support to theoretical predictions based on first-principles calculations, and indicates the important role played by electron–phonon interaction in phonon transport. It should be noted that although our measurement is dynamic, the measured electron-scattering properties of phonons should be equivalent to those in a doped semiconductor at thermal equilibrium, due to the fact that the photo-excited carriers have cooled down to a quasi-Fermi-Dirac distribution[47] with the same temperature as the lattice during the time window of our measurement.

The impact of this work is not limited to energy materials, such as thermoelectrics or photovoltaics. In any system where electron–phonon interactions play an important role, this technique can be used to extract phonon-specific information of electron–phonon interaction. An important case is the superconductors, especially high-temperature superconductors, to which we believe this experimental technique can be readily applied[48,49]. This technique can be further advanced by tuning the frequency of the coherent acoustic wavepackets[50] and by extending the method to transverse phonons[51], which will allow to measure the electron-scattering time of phonons with different frequencies and polarizations. Combined with the recent advances in the generation of coherent phonons above 1 THz in frequency[52], this technique can be a useful tool for analysing phonon scattering at frequencies comparable to those of heat-carrying thermal phonons at room temperature.

## Methods

**Details on experimental set-up.** We use an amplified Ti:sapphire laser system producing 300 fs pulses at 780 nm at a repetition rate of 250 kHz. The output of the laser system is split into a pump beam and a probe beam. The pump beam is subsequently modulated by an acousto-optic modulator (AOM) at a modulation frequency of 95 kHz. After passing through a frequency-doubling crystal (bismuth borate), part of the 780 nm photons are converted into 390 nm photons, and subsequently separated from the remaining 780 nm beam by a dichroic mirror. The two beams are then focused onto the sample with lenses. The extra path of the 780 nm beam with respect to the 390 nm beam is ~8 cm, corresponding to a delay of ~260 ps. Two thin-film polarizers are placed in the path of the 780 nm beam to control its power without deflecting the beam. The diameters of both beams at the sample plane (also the focal planes of the corresponding lenses) are measured with a razor-blade beam profiler to be 60 μm at $1/e^2$ intensity level. The probe beam goes through a delay stage and is frequency-doubled via a bismuth borate crystal. The 390 nm probe is then focused onto the sample with a lens (beam diameter 20 μm at $1/e^2$ intensity level). The reflected probe beam from the sample is directed into a photodiode via a beam splitter, which is then read-out by a lock-in amplifier at the modulation frequency of the AOM (95 kHz). The typical power for the 390 nm pump is 13 mW and for the 390 nm probe is 0.5 mW.

**Estimation of carrier concentration.** At 250 kHz repetition rate, 10 mW of measured laser power translates to single-pulse energy of 80 nJ (a factor of two takes into account the square-wave modulation of the AOM). The absorptance of the membrane at 780 nm is measured to be 25–30% depending on location. Given the photon energy of 1.59 eV at 780 nm, and the beam diameter of 60 μm, the generated concentration of electron–hole pairs can be estimated to be $1.8 \times 10^{19}$ cm$^{-3}$.

**Calculating phonon-scattering rate.** The damping of a coherent phonon mode when photo-excited carriers are present can be described by

$$A_2 = A_1 \varepsilon \exp\left[-\frac{1}{2\tau_{ep}}\Delta t\right],\qquad(2)$$

where $A_1$ and $A_2$ are the amplitudes of the first and second pulses, $\Delta t = 360$ ps is the duration of the time window when charge carriers are excited and $\varepsilon$ is the attenuation due to phonon–phonon interaction and boundary scattering. Here the factor of two inside the exponent accounts for the fact that the energy of an

acoustic pulse is proportional to the square of its amplitude. From equation (2) the scattering rate due to electron–phonon interaction $\gamma_{ep}$ can be calculated as

$$\gamma_{ep} \equiv \frac{1}{\tau_{ep}} = \frac{2}{\Delta t}\left[\ln\left(\frac{A_1}{A_2}\right) - \ln\left(\frac{A_1^0}{A_2^0}\right)\right],\qquad(3)$$

where $A_1^0$ and $A_2^0$ refer to the amplitudes of the two pulses without the 780 nm excitation.

**Data availability.** The data that support the findings of this study are available from the corresponding authors on request.

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

## Acknowledgements

We thank Hyun D. Shin for help with the laser system, and Jiawei Zhou and Yangying Zhu for helpful discussions. This article is based on work supported by S$^3$TEC, an Energy Frontier Research Center funded by the U.S. Department of Energy, Office of Basic Energy Sciences, under Award No. DE-FG02-09ER46577.

## Author contributions

B.L., A.A.M. and G.C. conceived this project; B.L. and A.A.M. did the experiment. All authors analysed the data, and B.L. and G.C. wrote the paper. All the authors read the paper and made comments. K.A.N. and G.C. supervised this project.

## Additional information

**Competing financial interests:** The authors declare no competing financial interests.

