## [Peer Review File · Nature Communications]

Reviewers' comments:

Reviewer #1 (Remarks to the Author):

This manuscript builds on previous work of the authors in which they managed to unravel the spectral contributions of phonons to the lattice thermal conductivity. Here, however, they characterized the contribution of electrons to the scattering of 250-GHz phonons in silicon. They used a three-laser technique with different optical wavelengths and a timing sequence that makes it possible to measure the scattering rate of the 250 GHz phonon when electrons are optically excited at the correct moment. I see no reason to doubt the experiment. The authors then carry out the measurement for various concentrations of the excited carriers, from 0.1 to 2.7×10^{19} per cubic cm (Fig. 4) and show a monotonic increase of the scattering rate as a function of concentration; this also makes sense. Better yet is the fact that this experimentally observed increase corresponds well to calculations based on a very simple deformation potential theory, with no adjustable parameters.

These results are quite surprising, since phonon scattering by conduction electrons is almost never considered in calculations of the lattice thermal conductivity. The surprise is justification enough for my opinion that the article is worth publishing in Nature Communications. The authors should discuss the following points.

1. How general is this result? The paper would benefit from identifying the dependence of the scattering rate not only on the electron concentration, but also on the phonon frequency and mode. If the authors have such data, I would encourage them to add them to this article rather than make a separate publication of it.
2. The data were obtained dynamically, with pumped electron-hole pairs. Is the scattering rate of phonons by electrons the same under static conditions? Do the electrons that exist at thermal equilibrium in a doped sample have the same effect? For example, optically pumped electrons could be surrounded by a polaron-like zone of deformation (as explained p. 8) which in itself is a phonon that could interact with other phonons anharmonically. If the electron distribution in the sample were uniform, this effect may not happen, and the phonon scattering by electrons might be very much weaker than that measured here.

The last comment is optional.

3. A nice additional experiment would be to verify Galilean invariance in this simple two-fluid model (electrons and phonons). If the reference frame for each particle in each population is the same, the interaction rate of band edge electrons in Si on the 240 GHz phonons should be the same as that of those phonons on those electrons. This can be written for the total electron and phonon population, but there the comparison makes little sense since either fluid interacts with a wide dispersion of particles in the other fluid; i.e. the relevant interactions can happen at various frequencies and the experiment is not very revealing. In the present experiment, the energies of both phonons and electrons are well defined, and this would be an interesting comparison. It does involve measuring the band edge electrons mobility scattering time (not lifetime). I am not sufficiently familiar with the experiment to know if this can be done, so this is only an optional suggestion, not linked to the acceptance of the manuscript. Perhaps an optical equivalent of the Shockley method to measure mobility could be invented and applied here?

Reviewer #2 (Remarks to the Author):

Authors report on experimental investigations of a single phonon mode interaction with electron and holes by measuring mode lifetime. For this study they developed an original technique based on three-pulse photoacoustic spectroscopy. The paper is clearly written and reported results are interesting. However theoretical interpretation of experimental data does not correspond to high quality criteria of Nature Communications:

- 1) Matthiessen's rule, used in the paper for the description of the phonon scattering, is not enough justified for the scattering of sub-THz phonons;
- 2) There are several theoretical works showing that for accurate description of phonon transport one should consider collective excitation instead of single-mode phonon (for example: Nano Lett. 14, 6109 (2014)).

These two technical issues may lead to erroneous interpretation of the results. Therefore I recommend to reject this manuscript and recommend to resubmit revised version in more specific journals, like as Physical Review Letters.

Revision Report NCOMMS-16-07310

Photo-excited Charge Carriers Suppress Sub-THz Phonon Mode in Silicon at Room Temperature

Bolin Liao, Alexei A. Maznev, Keith A. Nelson and Gang Chen

We highly appreciate the reviewers for their insightful comments and criticism, which have greatly helped us improve both the content and the presentation of our work. In particular, we thank both reviewers for commenting positively on the originality of our experimental technique, and the significance/interest of our experimental results. Especially the 1st reviewer points out the significance of the “surprise” factor of our results. The theory of the effect of electron-phonon interaction on phonons was first developed by Ziman in 1950s, and the earliest experiments dated back to the early 1960s. In the past 60 years, this effect was always regarded as negligible at room temperature, and researchers usually do not take it into account at all, especially in the literature of thermoelectrics. Even before we did the calculation and the experiment, we ourselves did not expect appreciable contribution of this effect. Our combined computation/experimental results, however, point out that our previous perception of this problem was flawed, and this effect must be correctly included when studying heavily-doped semiconductors in the future, a conclusion with impact on the whole field of semiconductor physics. We thus agree with the first reviewer that this “surprise” factor is the core contribution of our work. At the same time, we appreciate the 2nd reviewer’s criticism on our theoretical interpretation of the experimental results, and accordingly we revised the theoretical interpretation to clarify the concerns pointed out by the reviewer. We list the revisions and replies to specific comments/questions of the reviewers as follows (changes also highlighted in the main text):

Reviewer #1:

Comment: This manuscript builds on previous work of the authors in which they managed to unravel the spectral contributions of phonons to the lattice thermal conductivity. Here, however, they characterized the contribution of electrons to the scattering of 250-GHz phonons in silicon. They used a three-laser technique with different optical wavelengths and a timing sequence that makes it possible to measure the scattering rate of the 250 GHz phonon when electrons are optically excited at the correct moment. I see no reason to doubt the experiment. The authors then carry out the measurement for various concentrations of the excited carriers, from 0.1 to 2.7×10^{19} per cubic cm (Fig. 4) and show a monotonic increase of the scattering rate as a function of concentration; this also makes sense. Better yet is the fact that this experimentally observed increase corresponds well to calculations based on a very simple deformation potential theory, with no adjustable parameters.

These results are quite surprising, since phonon scattering by conduction electrons is almost never considered in calculations of the lattice thermal conductivity. The surprise is

justification enough for my opinion that the article is worth publishing in Nature Communications.

Response: We thank the reviewer for the accurate summary of our work, and especially for his/her positive comments on the originality and quality of this work and appreciation of the importance of the findings.

Comment: 1. How general is this result? The paper would benefit from identifying the dependence of the scattering rate not only on the electron concentration, but also on the phonon frequency and mode. If the authors have such data, I would encourage them to add them to this article rather than make a separate publication of it.

Response: In our *ab initio* simulation (ref. 33), we indeed calculated and reported the scattering rates due to electron-phonon interaction of all phonon modes in silicon. We found that for acoustic phonons near the zone center, the linear scaling of the scattering rate with phonon frequency is in general valid, while for other phonons modes the dependence is more complicated. In the experiment, however, there are a few technical issues that limit our current ability to access other phonon modes. First of all, the Brillouin frequency, which determines the phonon frequency we measure, only depends on the refractive index of the material and the wavelength of the probe pulse. Therefore, for a given material, we need a laser system with a widely tunable range of wavelength to measure phonons with different frequencies. It is currently not available in our laboratory but this measurement could be potentially done in the future. Secondly, it has been very challenging to generate coherent phonon modes with frequencies above 1 THz. In fact some recent works in our group have demonstrated generating above-THz phonons in superlattice structures, but so far there is no known way to generate them in an arbitrary sample. We added a paragraph in the main text to discuss these technical challenges and possible future directions of this technique on page 13: “This technique can also be augmented by coupling with ultrafast lasers with a tunable-wavelength, to be able to measure the electron-scattering time of phonons with different frequencies. Combined with the recent developments of THz acoustic wave generation methods [citation], this technique is envisioned to be a useful tool for analyzing scattering properties of thermal phonons”.

Comment: 2. The data were obtained dynamically, with pumped electron-hole pairs. Is the scattering rate of phonons by electrons the same under static conditions? Do the electrons that exist at thermal equilibrium in a doped sample have the same effect? For example, optically pumped electrons could be surrounded by a polaron-like zone of deformation (as explained p. 8) which in itself is a phonon that could interact with other phonons anharmonically. If the electron distribution in the sample were uniform, this effect may not happen, and the phonon scattering by electrons might be very much weaker than that measured here.

Response: We thank the reviewer for pointing out this important question. One possible difference between photo-excited and static carriers is their distribution functions. Static electrons or holes at thermal equilibrium in a doped sample assume a Fermi-Dirac

distribution. For photo-excited hot carriers, they first equilibrate within themselves through electron-electron interactions in ~ 100 femtoseconds, after which quasi Fermi-Dirac distributions are established with a higher temperature than the lattice. Then the carriers cool down through interaction with the lattice, and usually reach the lattice temperature within tens of picoseconds in silicon. Thus for our measurement, which utilizes the time window of ~ 200 ps to ~ 700 ps, the distributions of electrons and holes are essentially the same as those under the static condition. In terms of the “polaron-like” effect pointed out by the reviewer, if we understand it correctly, it is the effect of the strain gradient induced by the nonuniform distribution of the photo-excited charge carriers. Indeed this strain gradient can scatter phonons, but this effect is small in this specific case for two reasons: 1. The strain gradient is along the radial direction. Since carriers are uniformly generated along the light path, the strain is also uniform in the axial direction, along which the coherent phonon propagates; 2. The length scale of the radial strain gradient is very different from the length scale of the phonon mode we probe. The length scale of the radial strain gradient is set by the size of the laser beam, which is on the order of ~ 50 micrometers, while the wavelength of the phonon mode we study is ~ 40 nanometers. And thus, for this phonon mode we study, the strain is essentially uniform within a range of tens of its wavelengths. We added a paragraph in the manuscript to clarify this equivalence on page 6: “It is worth mentioning that, although our measurement is dynamic, the measured electron-scattering properties of phonons should be equivalent to those in a doped semiconductor at thermal equilibrium, due to the fact that the photo-excited carriers have cooled down to a quasi-Fermi-Dirac distribution [citation] with the same temperature as the lattice during the time window of our measurement. Furthermore, the additional strain gradient induced by the photo-induced carriers does not significantly alter the lifetime of the phonon mode we study in this experiment, for the following two reasons: 1) the strain gradient is along the radial direction, while the coherent phonon mode travels along the axial direction; 2) the length scale of the radial strain gradient (set by the laser beam size ~ 50 μ m) is much larger than the wavelength of the coherent phonon mode (~ 40 nm), and thus the strain seen by the coherent phonon mode is essentially uniform.”

Comment: The last comment is optional.

3. A nice additional experiment would be to verify Galilean invariance in this simple two-fluid model (electrons and phonons). If the reference frame for each particle in each population is the same, the interaction rate of band edge electrons in Si on the 240 GHz phonons should be the same as that of those phonons on those electrons. This can be written for the total electron and phonon population, but there the comparison makes little sense since either fluid interacts with a wide dispersion of particles in the other fluid; i.e. the relevant interactions can happen at various frequencies and the experiment is not very revealing. In the present experiment, the energies of both phonons and electrons are well defined, and this would be an interesting comparison. It does involve measuring the band edge electrons mobility scattering time (not lifetime). I am not sufficiently familiar with the experiment to know if this can be done, so this is only an optional suggestion, not linked to the acceptance of the manuscript. Perhaps an optical equivalent of the Shockley method to measure mobility could be invented and applied here?

Response: This is indeed a very interesting proposal. If we understand it correctly, here the Galilean invariance actually translates to the detailed balance principle: the transition rate of one process must be the same as its inverse process given the same population distributions, and this touches upon potentially an Onsager relation between phonon-limited electron transport and electron-limited phonon transport. For this measurement to be done, we need a technique to measure the phonon-limited mobility of a single electronic state. Indeed the Haynes-Shockley technique might be a good starting point, but as we see it, the main obstacles here include the need for an excitation source that only selectively excites a specific electron state, and a sensitive and sufficiently fast (same order as the electron scattering time) electrical detection mechanism to measure the transient response caused by the excitation. Given the recent development of single-photon sources and ultrafast oscilloscopes, this experiment might not be entirely impossible to do. We thank the reviewer for sharing this thought, and this might be a very interesting direction to pursue in the future.

Reviewer #2

Comment: Authors report on experimental investigations of a single phonon mode interaction with electron and holes by measuring mode lifetime. For this study they developed an original technique based on three-pulse photoacoustic spectroscopy. The paper is clearly written and reported results are interesting.

Response: We thank the reviewer for positive comments on the originality of our experimental technique, the presentation quality of our paper and the interest of our results.

Comment: However theoretical interpretation of experimental data does not correspond to high quality criteria of Nature Communications:

Response: We respectfully disagree with the reviewer on the statement that our theoretical interpretation of experimental data does not meet high standards. Our theoretical model is based on state-of-the-art first-principles simulations of electron-phonon interactions, utilizing DFT and Wannier-function interpolation. The theoretical work itself is highly nontrivial and in fact has been published in Physical Review Letters (Ref. 33). We understand the reviewer has specific concerns of the model, which we will address in the following sections, but we feel these concerns are relatively minor compared to the overall quality of our theoretical calculation. In addition, our theoretical model explains our experimental data reasonably well, and therefore we believe it serves its role in an experiment-oriented work.

Comment: 1) Matthiessen's rule, used in the paper for the description of the phonon scattering, is not enough justified for the scattering of sub-THz phonons;

Response: We appreciate the reviewer for pointing this out, and we agree with the reviewer that the use of the term "Matthiessen's rule" in our paper is not entirely appropriate. In fact the use of the term is not necessary either. Matthiessen's rule

concerns macroscopic properties such as electrical resistivity or thermal conductivity, and for those properties there are many examples when Matthiessen's rule is invalid. But when we discuss adding scattering rates due to electrons and holes on page 11, this is not really Matthiessen's rule: we are adding probabilities of independent scattering processes (In the revised manuscript, we clearly stated the assumption of independent scattering by electrons and holes in the manuscript. This assumption is valid in silicon since the electron-hole interaction, or exciton effect, is weak due to low exciton binding energy). There is no need to invoke Matthiessen's rule to support the basics of the probability theory. In the other case when Matthiessen's rule is mentioned (Eq. (2)), it is not necessary either, since we are measuring the attenuation due to the electron-phonon scattering on top of the attenuation due to other factors. Recognizing this flaw, we removed the use of the term "Matthiessen's rule" in both occasions (on page 10, we changed the original sentence "we use Matthiessen's rule to separate the contributions from electron-phonon interaction and phonon-phonon interaction", to "we separate the additional contribution of phonon attenuation from electron-phonon interaction by comparing the overall attenuations with and without the excitation beam"; on page 11, we changed the original sentence "The contributions from the electrons and holes are calculated independently and combined using Matthiessen's rule", to "The contributions from the electrons and holes are calculated separately and directly summed with the assumption that the phonon scattering due to electrons and holes are independent. This assumption is valid in silicon at room temperature given its low exciton binding energy [citation]", and we rewrote Eq. (2) to avoid the confusion, where we denote all other attenuation factors with a single factor ϵ except for the electron-phonon contribution.

Comment: 2) There are several theoretical works showing that for accurate description of phonon transport one should consider collective excitation instead of single-mode phonon (for example: Nano Lett. 14, 6109 (2014)).

Response: We are aware that the simple picture of phonon transport as the sum of single-phonon modes is not valid for *certain special cases*, such as graphene (discussed in the paper referred by the reviewer [Nano Letters, 14, 6109 (2014), referred to as NL in the following]) and diamond, where normal processes are strong. It is also known that it is in general invalid at low temperatures, for the same reason. But we want to emphasize that the single-mode picture *is valid in most materials* at room temperature for all practical purposes, as demonstrated by the excellent agreement between experiment and first-principles calculation of the thermal conductivity of numerous materials, published by different groups. In particular it is valid in silicon at room temperature (Esfarjani et al., PRB, 84, 085204 (2011)). Moreover, even in the special cases when the single-mode picture is not entirely valid, which means the single mode relaxation time approximation fails at calculating the thermal conductivity, the information of the lifetime of a single phonon-mode is still useful and actually necessary, because the method used in [NL] and other works to overcome this limitation still requires the knowledge of scattering rates of elementary scattering events involving single-mode excitations (such as 3-phonon scattering in [NL]). Extending the framework developed by Broido and used by [NL] and others onto the materials in which the contribution of electron-phonon scattering to thermal conductivity is important will, likewise, require the knowledge of scattering rates

for elementary processes involving single phonon mode excitations. Therefore measuring single-mode lifetimes still provides important information for understanding thermal transport, even in these special cases.

Comment: There two technical issues may lead to erroneous interpretation of the results. Therefore I recommend to reject this manuscript and recommend to resubmit revised version in more specific journals, like as Physical Review Letters.

Response: We respect the judgment and expertise of the reviewer, but we hope the clarifications provided above can at least earn a re-evaluation from the reviewer on our work. In fact we appreciate the reviewer's suggestion that we resubmit our manuscript to Physical Review Letters, an equally prestigious journal, which in our view shows his/her appreciation of the value of our work. In an era of growing open access journals, we believe potentially impactful works could reach a wider audience through publication in open access journals, and thus we think Nature Communications is a better place for our manuscript.

REVIEWERS' COMMENTS:

Reviewer #1 (Remarks to the Author):

Having reviewed the comments of both referees, and the responses made by the authors to them, I think this manuscript is now acceptable for publication in Nature Communications.

In my previous review, I expressed the opinion that the role of electron scattering of phonons in the lattice thermal conductivity is surprising, but properly put in evidence by the experiments described in this manuscript; this remains, the main reason I support publication. I had asked two questions: (1) can the result be made mode and frequency specific? And (2): is the result applicable to charge carriers in thermal equilibrium? The authors answered the questions and modified the paper in ways that I accept. The answer to the first question is that the necessary instrumentation is not yet available. The answer to the second question is that yes, the results obtained with optically pumped charge carriers should be equivalent to those expected for carriers induced by doping, for reasons outlined in the revised manuscript.

The paper would become more useful to experimentalists who do not have access to extended numerical modeling techniques if a simple semi-quantitative equation could be developed for phonon scattering frequencies by electrons, as a function of phonon frequency and temperature. I am asking for an expression suitable for use in a Callaway-type model. If the authors feel that this is not realistic, this request does not need to delay publication.

The comments of the second reviewer are, I think, addressed correctly. Reviewer #2 had two related objections: (1) the treatment of scattering events as being additive, and therefore independent of each other; and (2) the treatment of phonons as single particles. Obviously, both are crude approximations, but they historically been quite successful in calculating lattice thermal conductivities. The authors reply to objection (1) is rather indirect: they simply clarify that they treat phonon scattering events independently of each other rather than calling that assumption "Matthiesen's rule". They reply to objection (2) by stating the historical argument I give above.

The manuscript's main goal is to report that electron interactions with phonons are much more important than previously thought; the use of approximations, even very crude ones, in the calculations does not detract from that main conclusion.

Reviewer #2 (Remarks to the Author):

Authors convincingly enough replied to my technical questions and properly improved the manuscript. Although the theoretical model used can be further improved, I think that authors explanations added in the revised manuscript make it suitable for publication in Nature Communications. Therefore I am glad to recommend the revised manuscript for publication.

Revision Report NCOMMS-16-07310A

Photo-excited Charge Carriers Suppress Sub-THz Phonon Mode in Silicon at Room Temperature

Bolin Liao, A. A. Maznev, Keith A. Nelson and Gang Chen

All changes are highlighted in the manuscript.

General Changes:

1. We added a relevant literature (ref. 31 in the revised version) and correspondingly a brief discussion;
2. We determined the wavelengths of lasers used in the experiment to more accurate values (390 nm and 780 nm).

Reviewer #1:

Comment: ...The paper would become more useful to experimentalists who do not have access to extended numerical modeling techniques if a simple semi-quantitative equation could be developed for phonon scattering frequencies by electrons, as a function of phonon frequency and temperature. I am asking for an expression suitable for use in a Callaway-type model. If the authors feel that this is not realistic, this request does not need to delay publication.

Response: In fact Eq. 1 in our manuscript is such an equation for longitudinal acoustic phonons. It is derived with the deformation potential approximation and simple dispersion relations for electrons and phonons and fitted to the first-principles simulation results. It produces a phonon relaxation time that can be directly used in the Callaway model. A similar equation for longitudinal optical phonons was also derived in Ref. 34, and we added a sentence in our manuscript to clarify: “A similar expression for longitudinal optical phonons is also given in Ref. 34. These expressions can be directly used in Callaway-type models for estimating the thermal conductivity”.